# Mechanistic Insights into Anti-Melanogenic Effects of Fisetin: PKCα-Induced β-Catenin Degradation, ERK/MITF Inhibition, and Direct Tyrosinase Suppression

**DOI:** 10.3390/ijms262311739

**Published:** 2025-12-04

**Authors:** Zin Zin Ei, Satapat Racha, Hongbin Zou, Pithi Chanvorachote

**Affiliations:** 1Department of Pharmacology and Physiology, Faculty of Pharmaceutical Sciences, Chulalongkorn University, Bangkok 10330, Thailand; hushushin@gmail.com (Z.Z.E.); satapatto@gmail.com (S.R.); 2Center of Excellence in Cancer Cell and Molecular Biology, Faculty of Pharmaceutical Sciences, Chulalongkorn University, Bangkok 10330, Thailand; 3Interdisciplinary Program in Pharmacology, Graduate School, Chulalongkorn University, Bangkok 10330, Thailand; 4College of Pharmaceutical Sciences, Zhejiang University, Hangzhou 310058, China; zouhb@zju.edu.cn; 5Faculty of Pharmacy, Silpakorn University, Nakhon Pathom 73000, Thailand; 6Sustainable Environment Research Institute, Chulalongkorn University, Bangkok 10330, Thailand

**Keywords:** fisetin, melanogenesis, MITF, tyrosinase, PKCα, β-catenin, degradation

## Abstract

Excessive melanin production causes hyperpigmentation disorders such as freckles, melasma, and age spots, affecting appearance and quality of life. Tyrosinase is the key enzyme controlling melanin synthesis, and natural compounds are being explored as effective tyrosinase inhibitors. Fisetin, a dietary flavonoid found in fruits and vegetables like grapes and onions, is known for its anti-inflammatory and anticancer properties, but its anti-melanogenic activity remains unclear. This study demonstrated that fisetin, up to 60 μM, is non-toxic and significantly decreases tyrosinase activity and melanin content in human melanoma cells. Mechanistically, fisetin activates PKCα, leading to phosphorylation and degradation of β-catenin, thereby downregulating MITF expression. Additionally, it activates ERK and AKT/GSK3β pathways, promoting ubiquitination and proteasomal degradation of MITF, resulting in reduced levels of tyrosinase, TRP-1, and TRP-2. The proteasome inhibitor MG132 confirmed that fisetin accelerates β-catenin and MITF degradation. Additionally, inhibition of the PI3K/AKT pathway by LY294002 or the ERK pathway by PD98059 reversed fisetin’s reduction of tyrosinase activity and melanin synthesis, further verifying the participation of these pathways. Computational docking integrated with deep learning-based CNN scoring revealed that fisetin interacts with PKCα, β-catenin, tyrosinase, and TYRP1. Collectively, these findings suggest that fisetin exerts multi-targeted inhibitory effects on melanogenesis, highlighting its potential as a therapeutic and cosmetic agent for hyperpigmentation.

## 1. Introduction

Melanin is a pigment produced in melanocytes located at the basal layer of the epidermis, where it is synthesized within melanosomes and subsequently transferred to keratinocytes. Beyond its role in pigmentation, melanin provides essential photoprotection by absorbing and scattering ultraviolet (UV) radiation, neutralizing free radicals, and reducing cellular damage from external stressors [1,2].

However, excessive melanin production or accumulation can lead to hyperpigmentation disorders such as freckles, melasma, age spots, and senile lentigines, which negatively affect appearance and quality of life [3]. Factors influencing skin pigmentation include melanin synthesis, transport, degradation, epidermal proliferation, and stratum corneum thickness. Since tyrosinase is the rate-limiting enzyme in melanogenesis, its regulation has become a primary target for therapies aimed at treating hyperpigmentation [4,5]. In recent years, natural compounds and plant extracts have emerged as promising tyrosinase inhibitors for use in cosmeceutical skin-lightening formulations [6,7].

Melanogenesis occurs in melanosomes and is primarily regulated by the tyrosinase family of enzymes, including tyrosinase, tyrosinase-related protein 1 (TRP-1), and tyrosinase-related protein 2 (TRP-2/DCT) [8]. Tyrosinase, a copper-containing rate-limiting enzyme, catalyzes the hydroxylation of tyrosine to L-DOPA and its subsequent oxidation to dopaquinone, a common precursor for both eumelanin and pheomelanin. TRP-2 converts dopachrome to DHICA, which is then processed by TRP-1 to form eumelanin, while pheomelanin arises through nonenzymatic polymerization of dopaquinone [9]. The expression of yrosinase family enzymes is transcriptionally regulated by MITF, which binds to specific promoter motifs, making MITF a key upstream regulator of melanogenesis. Enhanced expression or activity of these enzymes significantly increases melanin production in melanocytes [10]. Tyrosinase, the rate-limiting enzyme that drives the initial step of melanin production, is considered a prime target for managing hyperpigmentation. However, effective melanin inhibitors must ensure long-lasting results while maintaining a favorable safety profile [11].

Protein kinase C (PKCα)-mediated β-catenin phosphorylation is a novel mechanism for regulating the Wnt/β-catenin pathway. Moreover, activation of PKCα induced the phosphorylation and ubiquitin proteasomal degradation of β-catenin [12,13]. A reduction in β-catenin levels leads to a corresponding decrease in MITF expression, as β-catenin functions as a key transcriptional coactivator that regulates the MITF gene [14,15].

Melanogenesis is regulated by multiple signaling pathways that modulate MITF stability and activity. The mitogen-activated protein kinase (MEK)/extracellular signal-regulated kinase (ERK) and phosphatidylinositol 3-kinase PI3K/protein kinase B (AKT)/glycogen synthase kinase-3β (GSK3β) pathways inhibit melanin production by promoting MITF degradation or preventing its binding to the tyrosinase promoter. Specifically, ERK activation suppresses melanogenesis by phosphorylating MITF, thereby targeting it for proteasomal degradation [16]. Activation of the JNK signaling pathway is also involved in downregulating the melanin-producing machinery, thereby leading to decreased melanin levels in melanoma cells [17]. The AKT signaling pathway regulates melanin production by phosphorylating GSK3β (Ser 9), which in turn promotes MITF degradation via the ubiquitin–proteasome system [18,19]. Therefore, targeting multiple regulators of MITF has been suggested as a new approach to achieve sustained hypopigmentation via suppression of tyrosinase expression.

Fisetin (3,7,3′,4′-tetrahydroxyflavone) is a flavanol found in various fruits and vegetables, including strawberries, grapes, apples, onions, and cucumbers, and is part of the flavonoid subclass of polyphenols [20]. Previous research has highlighted fisetin as a potential functional food component due to its antioxidant, anti-inflammatory, neuroprotective, anti-arthritic, anti-allergic, and anticancer properties, with reported effects against cancers such as lung, bladder, breast, prostate, colon, and pancreatic cancer [21].

Despite these benefits, the role of fisetin in melanogenesis remains controversial. Takekoshi et al. observed that certain flavonoids, including fisetin, can increase melanin content and tyrosinase activity in human melanoma cells [22] and that fisetin enhances melanogenesis in B16F10 cells via activation of the β-catenin signaling pathway [23]. Conversely, Shon et al. reported that fisetin suppresses α-MSH-induced intracellular and extracellular melanin production in murine B16F10 melanoma cells [24]. Other findings showed that the inhibitory action of fisetin on the β-catenin/MITF signaling pathway suggests its potential as an effective agent against 451Lu human melanoma cells [25].

In the present study, we revealed that the multi-target mechanism of fisetin inhibits melanogenesis in human melanoma cells by promoting the degradation of β-catenin via activation of PKCα, leading to a reduction in MITF expression and stability through activation of the ERK signaling pathway.

## 2. Results

### 2.1. Fisetin Reduces Melanin Synthesis in Human Melanoma Cells

The chemical structure of fisetin is illustrated in Figure 1A. Prior to examining its anti-melanogenic activity, the cytotoxic profile of fisetin was assessed in human melanin-producing cells. Cell viability was assessed by MTT assay in human melanoma cells treated with different concentrations of fisetin (0–80 μM) for 24 h. As shown in Figure 1B, fisetin exhibited no apparent cytotoxicity up to 60 μM, while a significant decrease in cell viability was observed at 80 μM. Based on these findings, subsequent experiments were conducted at concentrations of up to 40 μM.

As presented in Figure 1C, preliminary data indicated that fisetin reduced melanin synthesis at concentrations of 20 and 40 μM. To further confirm this effect, melanoma cells were pretreated with α-MSH (1 μM) for 24 h prior to fisetin exposure (0–40 μM). The results demonstrate that fisetin treatment markedly decreased melanin production in α-MSH-stimulated human melanoma cells.

Figure 1D shows that treatment with 20 and 40 μM fisetin for 24 h significantly decreased cellular melanin content. In particular, 40 μM fisetin reduced melanin synthesis by approximately 50-fold in human melanoma cells. Moreover, fisetin treatment (20, 40 μM) significantly suppressed melanin production in α-MSH-pretreated cells compared with α-MSH alone, confirming the potent anti-melanogenic effect of fisetin in human melanoma cells.

### 2.2. Reduced Cellular Tyrosinase Activity, Melanogenesis-Related Proteins, and mRNA Level in Fisetin-Treated Human Melanoma Cells

Since tyrosinase serves as the key enzyme in the melanogenesis process, the effect of fisetin on tyrosinase activity was assessed using both cell-free and cell-based assays. The indirect impact of fisetin on tyrosinase activity was evaluated using a cell-based assay, where equal amounts of protein extracted from fisetin-treated melanoma cells were used to catalyze the conversion of L-DOPA to dopachrome. As shown in Figure 2A, tyrosinase activity was markedly reduced in protein lysates obtained from cells treated with fisetin (20, 40 μM), both in untreated melanoma and α-MSH-stimulated melanoma cells. Specifically, cellular tyrosinase activity decreased by approximately 50-fold compared with untreated melanoma cells. Additionally, in α-MSH-induced melanoma cells, fisetin treatment led to about a 50-fold reduction in tyrosinase activity relative to the α-MSH group.

Based on previous findings demonstrating the direct inhibitory effect of flavonoids on tyrosinase activity [26], we further examined whether fisetin directly inhibits tyrosinase. Fisetin (0–40 μM) was added to a mixture containing tyrosinase enzyme extracted from human melanoma cells and its substrate, L-DOPA. After incubation at 37 °C for 2 h, dopachrome formation significantly decreased compared with control melanoma cells (Figure 2B). These results indicate that fisetin primarily reduces intracellular tyrosinase levels in human melanoma cells rather than acting solely as a direct enzyme inhibitor.

To investigate whether the reduction in melanin content and tyrosinase activity in fisetin-treated human melanin-producing cells was associated with changes in key melanogenic enzymes, we analyzed the expression levels of MITF and tyrosinase in human melanoma cells treated with different concentrations of fisetin (0–40 μM) for 24 h. The reduction in melanin production in fisetin-treated melanoma cells was evaluated at the protein level using immunofluorescence and Western blot analyses and at the mRNA level using qRT-PCR. As shown in Figure 2C, the fluorescence intensity corresponding to MITF and tyrosinase was markedly decreased in cells treated with fisetin for 24 h. Similarly, co-treatment with α-MSH and fisetin also led to a noticeable reduction in MITF and tyrosinase expression in melanoma cells.

To further validate the effect of fisetin on melanin synthesis, Western blot analysis was conducted to examine melanin-related protein expression in cells treated with fisetin, both in the presence and absence of α-MSH. The results demonstrated that fisetin treatment led to a marked reduction in MITF and tyrosinase protein levels in both melanoma and α-MSH-stimulated human melanoma cells (Figure 2D). Consistently, the mRNA expression of MITF-regulated melanogenic genes, including tyrosinase, TRP-1, PMEL, and TRP2/DCT, was significantly downregulated in fisetin-treated melanoma cells (Figure 2E,F).

### 2.3. Fisetin Inhibited Melanogenesis Through the Activation of the PKCα Pathway for Induced β-Catenin Degradation

PKCα promotes β-catenin phosphorylation, leading to its ubiquitin-proteasome-mediated degradation of β-catenin [12,13]. Figure 3A demonstrates that immunofluorescence analysis revealed a pronounced activation of PKCα in fisetin-treated cells (20, 40 μM) compared with both melanoma and α-MSH-stimulated melanoma cells. This activation of PKCα was accompanied by a marked decrease in the transcription factor β-catenin in fisetin-treated melanoma cells.

Western blot analysis further confirmed these findings, demonstrating that fisetin (40 μM) treatment elevated the expression of PKCα by approximately 3-fold in both melanoma cells and α-MSH-stimulated melanoma cells. Notably, the expression of β-catenin was dramatically reduced by approximately 10-fold in fisetin (40 μM)-treated melanoma cells and by 8-fold in α-MSH-stimulated cells (Figure 3B).

To examine this mechanism, the degradation of β-catenin in fisetin-treated melanoma cells was analyzed using an immunoprecipitation assay. Since β-catenin is a key transcriptional regulator upstream of MITF, we further examined whether fisetin influences β-catenin stability through the same degradation mechanism. Immunoprecipitation analysis revealed that fisetin (40 μM) treatment led to a 4-fold increase in the β-catenin–ubiquitin complex in human melanoma cells. Similarly, α-MSH-stimulated melanoma cells treated with fisetin showed a substantial increase in β-catenin ubiquitination. To confirm that this degradation occurred via the proteasomal pathway, melanoma cells were pretreated with MG132 (5 μM), a potent and reversible proteasomal inhibitor, for 1 h prior to fisetin exposure (40 μM) for 12 h. The results demonstrated that fisetin treatment significantly promoted proteasomal degradation of β-catenin in melanoma cells (Figure 3C).

### 2.4. Fisetin Suppressed Melanogenesis Through the ERK Signaling Pathway, Leading to a Reduction in MITF Degradation

To explore the upstream signaling mechanisms underlying the anti-melanogenic effect of fisetin, we examined the phosphorylation status of ERK and JNK. As shown in Figure 4A, immunofluorescence analysis revealed a pronounced increase in the phosphorylation of ERK and JNK in fisetin-treated cells (20, 40 μM) compared with both melanoma and α-MSH-stimulated melanoma cells. Fisetin treatment of melanoma cells resulted in a 2-fold increase in JNK phosphorylation, which promotes MITF degradation. This effect was further confirmed in α-MSH-stimulated melanoma cells, where fisetin treatment also enhanced JNK phosphorylation, leading to a reduction in melanin synthesis (Figure 4A). Moreover, the enhanced phosphorylation of ERK was accompanied by a marked induction of MITF degradation in both human melanoma and α-MSH-induced melanoma cells.

Western blot analysis further confirmed these findings, showing that fisetin treatment elevated ERK and JNK protein expression by approximately 5-fold and 3-fold, respectively, in melanoma cells. Similarly, the phosphorylation ratios (pERK/ERK and pJNK/JNK) were significantly increased in α-MSH-treated melanoma cells exposed to fisetin (Figure 4B).

To examine this mechanism, activation of ERK-induced degradation of MITF in fisetin-treated melanoma cells and α-MSH-induced melanoma cells was analyzed using an immunoprecipitation assay to detect ubiquitin-tagged MITF complexes. The results showed that treatment with fisetin (40 μM) markedly enhanced MITF ubiquitination, with a 3-fold increase in the MITF–ubiquitin complex compared with control cells. To confirm that this degradation occurred via the proteasomal pathway, cells were pretreated with MG132 (5 μM), a potent and reversible proteasome inhibitor, for 1 h prior to fisetin exposure (40 μM) for 12 h. The results demonstrated that fisetin treatment significantly promoted proteasomal degradation of MITF in both melanoma and α-MSH-induced melanoma cells by 2.5-fold and 5-fold, respectively (Figure 4C).

To verify the involvement of ERK signaling in fisetin-mediated melanogenesis inhibition, cells were co-treated with PD98059, a specific ERK inhibitor, together with fisetin. The results demonstrated that both melanin content and tyrosinase activity were substantially restored in the co-treatment group compared with cells treated with fisetin alone (Figure 4D,E).

Consistently, Western blot analysis demonstrated that co-treatment with fisetin and PD98059 reversed the fisetin-induced suppression of tyrosinase expression (Figure 4F). Moreover, the combined treatment significantly reduced the fisetin-induced elevation in ERK phosphorylation, suggesting that the ERK signaling pathway is critically involved in mediating fisetin’s anti-melanogenic activity (Figure 4G). As illustrated in Figure 4H, heat map analysis showed a marked decrease in the pERK/ERK phosphorylation ratio in melanoma cells co-treated with fisetin and PD98059. Consistently, immunofluorescence analysis revealed that co-treatment restored tyrosinase expression, effectively reversing the inhibitory effect observed with fisetin alone. Overall, fisetin reduces melanogenesis via activation of the ERK pathway by degradation of MITF.

### 2.5. Fisetin Inhibited Melanogenesis Through the Activation of the AKT/GSK3β Signaling Pathway

Phosphorylation of GSK3β by the PI3K/Akt signaling cascade has been reported to promote MITF degradation, thereby suppressing melanogenesis [27]. As illustrated in Figure 5A, immunofluorescence analysis revealed a remarkable increase in the phosphorylation levels of PI3K, Akt, and GSK3β in fisetin-treated cells (20, 40 μM), compared with both melanoma and α-MSH-stimulated melanoma cells. This enhanced phosphorylation of GSK-3β was accompanied by a pronounced reduction in MITF in both cell groups, indicating downstream suppression of melanogenic signaling.

Western blot analysis further confirmed these findings, demonstrating that fisetin treatment elevated the expression of phosphorylated PI3K, Akt, and GSK3β by approximately 1.2-, 2-, and 2-fold, respectively, in melanoma cells. Similarly, the phosphorylated to total protein ratios (pPI3K/PI3K, pAKT/AKT, and pGSK3β/GSK3β) were significantly increased in α-MSH-stimulated cells treated with fisetin (Figure 5B).

To elucidate the role of the PI3K/Akt/GSK3β signaling axis in fisetin-mediated suppression of tyrosinase, cells were pretreated with the PI3K-specific inhibitor LY294002 (10 μM) for 30 min prior to fisetin (40 μM) exposure for 24 h. As shown in Figure 5C,D, the inhibitory effects of fisetin on melanin synthesis and tyrosinase activity were notably attenuated by co-treatment with LY294002, suggesting the involvement of the PI3K/Akt pathway.

Consistent with these observations, Western blot analysis revealed that co-treatment reversed the fisetin-induced downregulation of tyrosinase expression (Figure 5E). Co-treatment with fisetin and LY294002 effectively attenuated the fisetin-induced increase in Akt phosphorylation, indicating that the PI3K/Akt signaling pathway plays a crucial role in mediating fisetin’s inhibitory effects on melanogenesis (Figure 5F). As shown in Figure 5G, the heat map revealed that pAKT/AKT phosphorylation levels were markedly reduced in melanoma cells co-treated with fisetin and LY294002. Furthermore, immunofluorescence analysis demonstrated that co-treatment restored tyrosinase expression, reversing the suppressive effect observed with fisetin treatment alone.

### 2.6. Molecular Docking of Fisetin with Target Proteins

After generating the docked poses, the next step was to quantitatively assess how closely they resembled the known reference structures. The standard metric for this comparison is the root-mean-square deviation (RMSD), which measures the average distance between corresponding atoms of two molecular structures. In redocking (cognate docking), a ligand is redocked into the receptor from which its bound pose was experimentally determined (Figure 6A–D). An RMSD below 2 Å for all redocked poses is generally considered indicative of a docking program’s ability to accurately reproduce the experimental binding conformation, reflecting the reliability of both the sampling algorithm and the scoring function.

For PKCα, fisetin exhibited two stabilizing hydrogen bonds within the ATP-binding pocket: the C4 carbonyl interacting with Val420 in the hinge region and the B-ring C3′-hydroxyl with Thr401. Among these, the Val420 contact represents the canonical hinge interaction reported for PKCα inhibitors such as staurosporine [28]. In terms of docking scores, fisetin showed a Vina score of –10.67 kcal/mol, which, although slightly lower than that of the co-crystallized inhibitor NVP-AEB071 (–12.13 kcal/mol), still indicates a strong binding affinity. The CNN pose scores of fisetin and NVP-AEB071 were 0.9265 and 0.9922, respectively, suggesting that the predicted binding orientation of fisetin was reliable and closely aligned with that of the reference inhibitor. Despite having a slightly lower CNN_VS value, fisetin demonstrates a well-stabilized conformation and favorable interactions within the ATP-binding site, supporting its potential as a PKCα inhibitor.

For β-catenin, fisetin showed CNN_VS and Vina scoring results that were not better than those of the β-catenin inhibitor R9Q. However, the binding pattern of fisetin and R9Q within the site was the same: in the loop region between armadillo repeats 2 and 3, fisetin formed a hydrogen bond with Ser246, as previously reported for R9Q [29]. These findings suggest that fisetin also has potential affinity for this site.

For tyrosinase, the docking analysis showed that the catechol moiety of fisetin lay coplanar with the tropolone scaffold of the reference inhibitor within the tyrosinase active site. Moreover, fisetin was predicted to form a hydrogen bond with Arg268, a residue repeatedly reported as essential for tyrosinase inhibition [30]. According to the Vina scoring function, fisetin exhibited a binding energy of –6.89 kcal/mol, which was more favorable than that of tropolone (−5.83 kcal/mol). The CNN_VS values of fisetin (3.5685) and tropolone (3.5963) were nearly identical, reflecting similar predicted binding quality. Fisetin showed a slightly lower CNN pose score (0.7317 vs. 0.9375) but a higher CNN affinity (4.877 vs. 3.836), suggesting that its slightly higher predicted binding affinity balances the modest difference in pose score. This balance between pose accuracy and affinity explains the close CNN_VS values for both ligands. Considering the overall binding energies, CNN metrics, and the key hydrogen bond with Arg268, these findings indicate that fisetin may exhibit a binding capability toward tyrosinase that is comparable to or even stronger than tropolone.

For TYRP1, fisetin formed a hydrogen bond with the water molecule bridging the two zinc ions and occupied the same binding site as kojic acid. Notably, the distance between fisetin and the bridging water molecule was approximately 2 Å, compared with about 3 Å for kojic acid (Figure 6). This observation suggests that fisetin adopts a more stable conformation within the binding site. Consistently, both the CNN_VS and Vina scoring results indicated that fisetin exhibits superior binding affinity relative to kojic acid.

## 3. Discussion

While melanin production protects against UV-induced DNA damage, prolonged UV exposure can lead to excessive melanin accumulation, causing various hyperpigmentation disorders [31]. Several strategies have been developed to identify bioactive compounds capable of preventing or alleviating skin hyperpigmentation disorders. Currently, widely used skin-whitening agents include kojic acid, arbutin, and niacinamide. However, these agents often exhibit limitations such as potential cytotoxicity and undesirable side effects. For instance, kojic acid has been associated with hepatocellular carcinoma, while prolonged use of arbutin may cause irreversible depigmentation. Therefore, the search for safer and more effective natural alternatives has become an important focus in the development of skin-lightening therapeutics [32]. Therefore, researchers have shown increasing interest in exploring natural products to identify compounds that can effectively reduce melanogenesis for use as skin-whitening agents.

The biological activity of flavonoids is closely associated with their molecular structure, particularly the number and position of functional groups such as hydroxyl moieties, as well as their ability to interact with biological membranes, the first natural barrier to cellular entry [33].

Fisetin is a biologically active flavone with a diphenylpropane backbone, consisting of three aromatic rings, four hydroxyl groups, and one keto (oxo) group [20]. Fisetin is widely distributed among various plant species and has attracted attention for its potent bioactivity. It has been recognized as a promising chemopreventive and chemotherapeutic agent in multiple types of cancer, as well as a neuroprotective compound against ischemia-induced brain damage, largely due to its strong antioxidant properties [20,34]. Although fisetin has been extensively studied for its pharmacological benefits in animal models of human diseases, its role in melanogenesis remains unclear, with studies reporting both inhibitory and stimulatory effects [22,24].

Previous studies have examined how the oxidative state regulates melanin synthesis, influencing the release of melanosomes in melanocytes [35]. Tyrosinase is a key enzyme in melanogenesis, catalyzing two critical reactions in the Raper–Mason pathway. First, through its activity, tyrosinase converts L-tyrosine into dopaquinone. Subsequently, dopaquinone is non-enzymatically transformed into dihydroxyphenylalanine (DOPA), a process accompanied by the generation of reactive oxygen species (O_2_^−^) [36]. In the second reaction, tyrosinase catalyzes the oxidation of L-DOPA to dopaquinone through its catecholase activity [37]. Specifically, tyrosinase, the rate-limiting enzyme in melanogenesis, catalyzes reactions that generate O_2_^−^ while producing both DOPA and dopaquinone [36]. These findings suggest that the balance between pro-oxidant and antioxidant states plays a critical role in regulating melanogenesis by modulating tyrosinase activity. Currently, many antioxidants have been shown to exhibit anti-melanogenic effects by suppressing tyrosinase activity and downregulating their upstream regulatory genes, including MITF [38]. As anticipated, fisetin, a potent antioxidant, inhibited vitro tyrosinase activity in the present study. Based on this observation, we hypothesized that fisetin acts as a negative regulator of melanogenesis in melanoma cells. Consistent with the present findings, Takekoshi et al. reported that certain flavonoids, including fisetin, can increase melanin content and tyrosinase activity in human melanoma cells, and that fisetin enhances melanogenesis in B16F10 cells via activation of the β-catenin signaling pathway [22]. In contrast, Son et al. demonstrated that fisetin reduced both intracellular and extracellular melanin levels in murine B16F10 melanoma cells [24].

Our results demonstrate that fisetin exhibits relatively low cytotoxicity up to 60 μM (Figure 1B) and effectively inhibits melanin production in both basal and α-MSH-stimulated conditions. At non-toxic concentrations (40 μM), fisetin significantly suppresses melanin synthesis in human melanoma cells (Figure 1C,D).

Fisetin significantly inhibited tyrosinase activity in both cell-free and cell-based assays. Treatment with 20 and 40 μM fisetin markedly reduced tyrosinase activity in melanoma and α-MSH-stimulated melanoma cells, showing approximately a 50-fold decrease compared with their respective control groups (Figure 2A). Fisetin treatment (0–40 μM) significantly reduced dopachrome formation in a tyrosinase L-DOPA reaction, suggesting some direct inhibitory effect on tyrosinase activity. However, the results indicate that fisetin mainly decreases intracellular tyrosinase levels in human melanoma cells rather than functioning solely as a direct enzyme inhibitor (Figure 2B).

Moreover, fisetin reduced the protein expression levels of MITF and tyrosinase in both human melanoma cells and α-MSH-stimulated melanoma cells (Figure 2C,D). MITF, a key transcription factor, regulates the expression of melanogenic enzymes, including tyrosinase, TRP-1, and TRP-2, and is controlled at both transcriptional and translational levels [39]. In addition, fisetin significantly decreased mRNA levels and suppressed the expression of MITF, tyrosinase, TRP-1, TRP-2, and PMEL in both melanoma and α-MSH-stimulated melanoma cells (Figure 2E,F).

PKCα, a conventional isoform of protein kinase C (PKC), functions as a well-established binding partner of β-catenin and promotes its degradation. Activation of PKCα induces phosphorylation of β-catenin at specific serine residues, thereby targeting it for ubiquitin-proteasome-mediated degradation [12,13].

PKCα plays a critical role in targeting β-catenin for proteasomal degradation via ubiquitination. In the absence of Wnt ligand binding, β-catenin is continuously phosphorylated by a destruction complex composed of Axin, adenomatous polyposis coli (APC), casein kinase 1 (CK1), and GSK3β. Sequential phosphorylation by CK1 and GSK3β generates a recognition site for the E3 ubiquitin ligase β-TrCP (β-transducin repeat-containing protein), which ubiquitinates β-catenin and directs it to the 26S proteasome for degradation. This ubiquitin–proteasomal degradation of β-catenin reduces MITF expression and stability, leading to downregulation of melanogenic enzymes. Consequently, melanin synthesis is significantly decreased, underscoring the pivotal role of β-catenin–MITF signaling in the regulation of pigmentation [12,13,14,15].

Fisetin activates PKCα in melanoma cells, leading to enhanced phosphorylation and ubiquitin–proteasome-mediated degradation of β-catenin. Immunofluorescence and Western blot analyses showed a strong increase in PKCα expression alongside a marked reduction in β-catenin, while immunoprecipitation confirmed that fisetin significantly increased β-catenin ubiquitination and promoted its proteasomal degradation, as verified by MG132 inhibition (Figure 3).

Multiple signaling pathways contribute to the downregulation of MITF, including the ERK and AKT/GSK3β pathways [40]. In the present study, fisetin exerted its anti-melanogenic effects by modulating the ERK and AKT/GSK3β signaling pathways. Consistent with previous reports, hypopigmentation is largely regulated through MAPK-mediated downregulation of MITF [41,42]. The present study revealed that fisetin markedly upregulated ERK phosphorylation, resulting in the suppression of tyrosinase activity. This involvement of the ERK pathway was further confirmed by co-treatment with the ERK inhibitor PD98059, indicating that ERK signaling is partially responsible for the hypopigmenting effects of fisetin (Figure 4). A similar anti-melanogenic mechanism has been reported for sulforaphane, which increased phospho-ERK levels while reducing phosphorylated p38, ultimately inhibiting melanogenesis [43].

Phosphorylated JNK may play a crucial role in inhibiting melanin production in melanoma cells, likely through negative regulation of the MITF signaling cascade. The selective JNK activator anisomycin suppresses melanogenesis via phosphorylation of CREB-regulated transcription coactivator 3 (CRTC3), whereas JNK inhibition enhances melanin synthesis by promoting CRTC3 dephosphorylation and nuclear translocation. In our findings, fisetin treatment in both melanoma cells and α-MSH-stimulated melanoma cells induces JNK phosphorylation, providing a supportive pathway for MITF degradation and consequent suppression of melanin production [17,44].

The PI3K/AKT/GSK3β pathway is another critical signaling cascade that regulates the transcriptional activity of MITF. Previous studies have demonstrated that activation of PI3K/AKT suppresses melanin accumulation in both murine melanocytes and human melanoma Melan-A cells [45]. In this study, fisetin increased the phosphorylation of Akt and GSK3β, resulting in reduced tyrosinase expression. The PI3K-specific inhibitor LY294002 was used as a negative control to validate the anti-melanogenic effects of fisetin (Figure 5).

Glycogen synthase kinase-3β (GSK3β) is a ubiquitously expressed and evolutionarily conserved kinase that regulates glycogen synthesis and modulates the expression of tau, a neuronal microtubule-associated protein [46,47]. GSK3β directly regulates MITF transcription, thereby controlling MITF-mediated melanin synthesis [48]. Recent studies have demonstrated that activation of the phosphatidylinositol 3-kinase (PI3K)/protein kinase B (Akt) pathway induces phosphorylation of GSK3β at Ser9, ultimately leading to downregulation of MITF expression.

The docking and CNN-based analyses provide mechanistic insight into how fisetin may exert its anti-melanogenic effect through interactions with key enzymes and signaling proteins involved in melanin synthesis. The redocking validation confirmed the reliability of the docking protocol, supporting the interpretability of the predicted binding modes.

For PKCα, fisetin formed two stabilizing hydrogen bonds within the ATP-binding pocket: the C4 carbonyl interacting with Val420 in the hinge region and the B-ring C3′-hydroxyl engaging Thr401. The hinge-directed interaction with Val420 is particularly significant, as this residue represents a conserved anchoring site frequently occupied by potent PKCα inhibitors such as staurosporine and maleimide analogs. Fisetin’s Vina score (–10.67 kcal/mol) and high CNN pose value (0.9265) further support a stable hinge-binding conformation. These results suggest that fisetin may competitively interact within the ATP-binding cleft of PKCα, potentially attenuating its kinase activity and downstream phosphorylation events. Because PKCα regulates MITF expression through β-catenin signaling [49,50], such modulatory activity could represent an upstream mechanism contributing to fisetin’s observed anti-melanogenic activity.

Although the docking results suggested that fisetin binds within the ATP-binding pocket of the PKCα-a region typically associated with ATP-competitive inhibitors such as staurosporine, our experimental data demonstrated an increase in PKCα phosphorylation and activity. This apparent discrepancy may be explained by the ability of flavonoids to modulate PKC signaling in a concentration-dependent manner. Earlier studies have reported that quercetin and related polyphenols exhibit biphasic effects on PKC activity, acting as activators at low concentrations and inhibitors at higher concentrations [51]. In addition, several reports have shown that polyphenolic compounds can regulate PKC activation through redox control, membrane association, or interaction with lipid cofactors, thereby maintaining or promoting kinase activation under specific cellular conditions rather than inhibiting it [52].

Therefore, even though fisetin binds at the ATP-binding pocket, it may stabilize PKCα in an active catalytic state or function as a modulator that facilitates its activation under the tested cellular context, leading to β-catenin phosphorylation, subsequent degradation, and suppression of MITF expression.

Although the binding affinity of fisetin toward β-catenin was not superior to that of the reference inhibitor R9Q, both ligands occupied the same loop region between armadillo repeats 2 and 3 and formed a hydrogen bond with Ser246, a residue implicated in β-catenin’s conformational regulation. This suggests that fisetin may interact within a regulatory loop region of β-catenin, potentially influencing its conformational stability or protein–protein interactions. Given that β-catenin signaling contributes to the transcriptional activation of MITF [14], such binding could provide a structural basis for fisetin’s experimentally reported ability to downregulate MITF and reduce melanogenesis.

For tyrosinase, fisetin adopted a binding orientation that closely overlapped with that of the reference inhibitor, tropolone, maintaining the characteristic interaction with Arg268, a residue widely recognized as essential for catalytic inhibition [50]. Although the CNN pose score of fisetin was slightly lower, its predicted affinity was marginally stronger, suggesting a comparable overall binding quality. This observation aligns with previous reports that polyphenolic compounds containing a catechol or resorcinol moiety can effectively chelate the dinuclear copper center and stabilize within the tyrosinase active site. Therefore, fisetin may suppress melanogenesis at the enzymatic level by directly interfering with tyrosinase activity.

In the case of TYRP1, fisetin was predicted to form a hydrogen bond with the bridging water molecule between the two zinc ions, exhibiting a shorter interaction distance than kojic acid. This configuration indicates a more favorable coordination geometry, consistent with its improved binding energy and CNN_VS score. Because TYRP1 functions cooperatively with tyrosinase during melanin polymerization [53], inhibition at this site could further enhance fisetin’s depigmenting potential through a synergistic effect on the melanogenic cascade.

Taken together, these findings propose that fisetin can interact favorably with PKCα, β-catenin, tyrosinase, and TYRP1 through stable binding conformations and conserved hydrogen-bonding motifs. This integrative computational evidence supports the hypothesis that fisetin’s anti-melanogenic action may involve both upstream suppression of MITF expression via PKCα/β-catenin signaling and downstream inhibition of enzymatic activity through direct interactions with tyrosinase and TYRP1 (Figure 6).

Overall, fisetin acts as a negative regulator of melanogenesis in human melanoma cells by lowering intracellular tyrosinase levels and inhibiting MITF expression. This effect is likely mediated through PKCα activation, which promotes β-catenin degradation, along with modulation of the ERK and AKT/GSK3β signaling pathways that contribute to MITF degradation, ultimately leading to reduced melanin synthesis (Figure 7).

## 4. Materials and Methods

### 4.1. Cell Culture

Human melanoma cells (G361) were obtained from the American Type Culture Collection (ATCC, Manassas, VA, USA). The accession numbers for G361 from the ATCC (Homo sapiens) are CVCL_1220. These cells were cultured in McCoy’s 5a medium (cat. no. 30-2007, ATCC, Manassas, VA, USA). This media was supplemented with 10% fetal bovine serum (FBS) (cat no: SV30160.03, HyClone™, Cytiva, Global life Sciences, New South Wales, Austria), 2 mM L-glutamine (Ref no: 35050-061, Gibco, Gaithersburg, MA, USA), and 100 units/mL of antibiotic–antimycotic (Ref no: 15240-062, Gibco, Grand Island, NY, USA), and the cells were incubated at 37 °C with 5% CO_2_ until they reached 70–80% confluence for subsequent experiments.

### 4.2. Reagents and Chemicals

The reagents 3-(4,5-dimethylthiazol-2-yl)-2,5-diphenyl tetrazolium bromide (MTT), bovine serum albumin (BSA), Hoechst 33342, synthetic melanin (cat no: M8631), and Dopa (cat no: 333786) were procured from Sigma-Aldrich, Co. (St. Louis, MO, USA).

Mouse monoclonal primary antibodies for MITF (cat no: ab12039) were obtained from Abcam (Waltham, MA, USA). Mouse monoclonal antibody to tyrosinase (cat no: SC20035) was procured from Santa Cruz Technology (Dallas, TX, USA).

Primary antibodies, including rabbit monoclonal antibodies for Akt (cat no: 9272S), p-Akt (Ser473, cat no: 4060S), mouse monoclonal antibodies for GSK3β (cat no: 9832S), rabbit monoclonal antibodies for GSK3β (cat no: 9323S), rabbit monoclonal antibodies for β-catenin (cat no: 8480S), monoclonal antibodies for PKCα (cat no: 2056S), rabbit monoclonal antibodies for JNK (cat no: 9252S), mouse monoclonal antibodies for pJNK (cat no: 9255S), rabbit monoclonal antibody to Erk1/2 (cat no: 4695S), rabbit monoclonal antibody to pErk1/2 (phosphorylated Thr202/Tyr204; cat no: 4370S), rabbit monoclonal antibody to GAPDH (cat no: 2118), anti-mouse IgG horseradish peroxidase (HRP)-linked secondary antibody (cat no: 7076S), and anti-rabbit IgG HRP-linked secondary antibody (cat no: 7074S) were bought from Cell Signaling Technology (Denver, MA, USA).

Secondary antibodies, including Alexa Fluor™ 594 goat anti-rabbit IgG (H + L) (cat no: A11037) and Alexa Fluor™ 488 goat anti-mouse IgG (H + L) (cat no: A11029), were supplied by Invitrogen, a division of Thermo Fisher Scientific (Carlsbad, CA, USA).

### 4.3. Preparation of Stock Solution for Fisetin

The fisetin was made into a 50 mM stock solution by dissolving it in dimethyl sulfoxide (DMSO) and was then stored at −20 °C. Under high treatment conditions, the final concentration of DMSO was 0.2% *v*/*v*, which did not exhibit any cytotoxic effects.

### 4.4. Cell Viability Assay

Cell viability was evaluated by employing the MTT assay. Human melanoma cells were seeded at a density of 1.5 × 10^4^ cells per well in 96-well plates and incubated for 48 h at 37 °C. The cells were then treated with various concentrations of fisetin (0–80 µM) for another 24 h. The following day, the fisetin-containing medium was removed and replaced with 4 mg/mL MTT reagent in PBS. After incubating for 3 h, 100 µL of dimethyl sulfoxide was added to dissolve the formazan crystals, and the absorbance of the formazan product was measured at 570 nm using a microplate reader.

### 4.5. Cellular Melanin Content Measurement

Human melanoma cells were plated in 6-well culture plates at a density of 3 × 10^5^ cells/well for 48 h. The following day, the cells were pretreated with melanin-stimulating hormone (α-MSH) (1 μM) for 24 h, followed by co-treatment with fisetin (0–40 μM) for a further 24 h. After that, the cells were washed with PBS, and the cell pellet was collected. To the cell pellet, 100 μL of working solution containing 1 M of NaOH and 10% DMSO was added, and the mixture was incubated at 80 °C for 3 h. After complete dissolution of the pellet, the absorbance at 405 nm was measured to assess the intracellular melanin content and converted to melanin content compared with the standard curve of synthetic melanin. The melanin content was normalized to the total amount of cellular protein and is presented as a relative value to untreated control cells.

### 4.6. Cell-Free Tyrosinase Activity Assay

The direct inhibitory effect on tyrosinase activity was evaluated according to previously established methods [54]. Human melanoma cells were cultured until 80% confluence then collected by trypsinization and centrifugation at 2500 rpm for 5 min at 4 °C. The cell pellets were lysed with 1% Triton X-100 in PBS containing 1% protease inhibitor at 4 °C with continuous vortexing every 10 min for 1 h. The cell lysate was centrifuged at 12,000 rpm for 15 min at 4 °C to obtain the supernatant. Various concentrations of fisetin (0–40 μM) prepared in PBS were added to the supernatant containing an equal amount of protein (50 μg). The mixture was incubated at 37 °C for 10 min, followed by the addition of 2 mM L-DOPA and a further 2 h of incubation at 37 °C. The OD of the forming dopachrome was measured via a microplate reader at 490 nm.

### 4.7. Evaluation of Cellular Tyrosinase Activity

The activity of tyrosinase in cellular samples was evaluated through a modified procedure derived from previously established techniques [55]. Human melanoma cells in 6-well plates (3 × 10^5^ cells/well) were cultured for 48 h and then treated with fisetin (0–40 μM) for 24 h. Then, the cells were collected by trypsinization and centrifuged at 2500 rpm for 5 min at 4 °C. Cell pellets were resuspended in 1% Triton X-100 prepared in PBS containing 1% protease inhibitor and vortexed every 10 min for 1 h at 4 °C. The lysates were then centrifuged at 12,000 rpm for 15 min at 4 °C, and the resulting supernatant was collected for protein quantification using a BCA assay kit. Equal amounts of protein (50 μg) from each sample were then mixed with 2 mM L-DOPA (50 μL) and incubated at 37 °C for 2 h. The absorbance of the resulting dopachrome was measured at 490 nm using a microplate reader.

### 4.8. Immunofluorescence

Human G361 cells were seeded at a concentration of 1.5 × 10^4^ cells per well in 96-well plates for 48 h and treated with melanin-stimulating hormone (α-MSH) (1 μM) for 24 h, followed by co-treatment with fisetin (0–40 μM) for a further 24 h. Following treatment, the cells were washed with PBS, fixed with 4% paraformaldehyde for 10 min, and permeabilized using 0.5% Triton-X in PBS for 5 min at room temperature. Non-specific proteins were blocked with 10% FBS in 0.1% Triton-X PBS for 1 h at room temperature. The cells were then exposed to a 1:400 dilution of primary antibodies and incubated overnight at 4 °C. The next day, the cells were washed with 10% FBS in 0.1% Triton-X and subsequently incubated with the secondary antibodies Alexa Fluor 594 conjugated goat anti-rabbit IgG (H + L) and Alexa Fluor 488 conjugated goat anti-mouse IgG (H +  L) along with Hoechst 33,342 for nuclear staining for 1 h at room temperature. After washing with PBS, the cells were fixed with 50% glycerol. Immunofluorescence images were captured using a fluorescence microscope (Olympus-IX51 with a DP70 digital camera, Olympus, Tokyo, Japan), and fluorescence intensity was analyzed using ImageJ software (ImageJ 1.52a, Rasband, W., National Institutes of Health, Bethesda, MD, USA).

### 4.9. Quantitative Analysis for Real-Time PCR Analysis

RNA extraction was conducted on cells treated with fisetin (3 × 10^5^ cells per well in 6-well plates) using GENEzol reagent. The total RNA extracted was utilized to synthesize complementary DNA (cDNA) using SuperScript III reverse transcriptase from Invitrogen. Reverse-transcription quantitative polymerase chain reaction (RT-qPCR) was performed with 100 ng of cDNA, utilizing Luna^®^ Universal qPCR Master Mix (NEB, Hitchin, UK) for a final reaction volume of 20 µL. The reactions were carried out using the CFX 96 Real-time PCR system from Bio-Rad in Hercules, CA，USA. The RT-PCR protocol consisted of an initial denaturation step at 95 °C for 1 min, followed by 45 cycles of denaturation at 95 °C for 15 s and primer annealing at 60 °C for 30 s. Melting curve analysis was performed to verify the specificity of the primers. The targeted genes of primers are:

MITF forward primer: 5′-TCATCCAAAGATCTGGGCTATGACT-3′

MITF reverse primer: 5′-GTGACGACACAGCAAGCTCAC-3′

tyrosinase forward primer: 5′-TCATCCAAAGATCTGGGCTATGACT-3′

tyrosinase reverse primer: 5′-GTGACGACACAGCAAGCTCAC-3′

TRP-1 forward primer: 5′-AAGGCTACAACAAAAATCACCAT-3′

TRP-1 reverse primer: 5′-ATTGAGAGGCAGGGAAACAC-3′

PMEL Forward: 5′-GTCAGCACCCAGCTTGTCA-3′

PMEL Reverse: 5′-GCTTCATTAGTCTGCGCCTGT-3′

Dct forward primer: 5′-GCAGCAAGAGATACACAGAAGAA-3′

Dct reverse primer: 5′-TCCTTTATTGTCAGCGTCAGA-3′

The PCR products were normalized using GAPDH genes as an internal control. The relative mRNA expression levels of the target gene were determined from the comparative Cq values.

### 4.10. Western Blot Analysis

Human melanoma cells were seeded in 6-well plates at a density of 3 × 10^5^ cells per well and cultured for 48 h. The cells were pretreated with α-MSH (1 μM) for 24 h, followed by co-treatment with varying concentrations of fisetin (0–40 μM) for an additional 24 h. After treatment, cells were harvested, washed with ice-cold PBS, and lysed on ice for 40 min in buffer containing 50 mM HEPES (pH 7.5), 150 mM NaCl, 5 mM EDTA, 1% Triton X-100, 1 mM PMSF, and 2 μg/mL pepstatin A, supplemented with a complete protease inhibitor cocktail (Roche, Basel, Switzerland, #04693116001). The lysates were centrifuged at 12,000× *g* for 15 min at 4 °C, and protein concentration was determined using a BCA protein assay kit (Thermo Fisher Scientific, Rockford, IL, USA). Equal amounts of protein were denatured at 95 °C for 5 min in sample buffer and separated by SDS-PAGE.

Proteins were then transferred onto a 0.45 μm nitrocellulose membrane, blocked with 5% non-fat milk in TBST (25 mM Tris-HCl, pH 7.5, 125 mM NaCl, 0.05% Tween-20) for 1 h, and incubated overnight at 4 °C with primary antibodies diluted 1:1000 in 5% BSA/TBST. GAPDH served as the loading control. After washing three times with TBST, membranes were incubated with HRP-conjugated anti-rabbit or anti-mouse secondary antibodies (1:2000 in 5% skim milk/TBST) for 1 h at room temperature. Protein bands were visualized using an enhanced chemiluminescent substrate (SuperSignal West Pico, Pierce, Rockford, IL, USA) and detected with the iBright™ CL1500 Imaging System (Invitrogen, Carlsbad, CA, USA, #A44240). Band intensity was quantified using ImageJ software (version 1.52a, NIH, USA).

### 4.11. Immunoprecipitation

Human melanoma cells pretreated with α-MSH and subsequently exposed to fisetin were harvested using cold PBS and lysed on ice in buffer containing 50 mM HEPES (pH 7.5), 150 mM NaCl, 5 mM EDTA, 1% Triton X-100, 1 mM PMSF, 2 μg/mL pepstatin A (Cell Signaling, #9803), and a complete protease inhibitor cocktail (Roche, #04693116001). The lysates were incubated on ice for 40 min and centrifuged at 12,000× *g* for 15 min at 4 °C to collect the supernatant.

Immunoprecipitation was performed using the Dynabeads™ Protein G Immunoprecipitation Kit (Thermo Fisher Scientific, Waltham, MA, USA). Magnetic beads were incubated with β-catenin or MITF antibodies (2 μL) in 100 μL of antibody-binding buffer for 15 min at room temperature with rotation. The bead–antibody complex was then mixed with the protein lysate and incubated overnight at 4 °C to allow for antigen binding. After washing three times with 200 μL of washing buffer, the bead–antibody–antigen complex was resuspended in 30 μL of lysis buffer, combined with 5 μL of 6 × sample buffer, and denatured at 95 °C for 5 min.

The samples were separated by 12% SDS-PAGE and transferred onto a 0.45 μm nitrocellulose membrane. The membrane was blocked with 5% non-fat dry milk in TBST (25 mM Tris-HCl, pH 7.4, 125 mM NaCl, 0.05% Tween-20) for 1.5 h and then incubated overnight at 4 °C with mouse anti-ubiquitin antibody (#14049). After washing, membranes were incubated with HRP-conjugated anti-mouse IgG secondary antibody (1:2000 in 5% skim milk/TBST) for 1 h at room temperature. Protein bands were visualized using an enhanced chemiluminescence (ECL) detection system and imaged on an iBright™ CL 1500 Imaging System. Band intensity was quantified using ImageJ software (version 1.52a, NIH, Bethesda, MD, USA).

### 4.12. Computational Analysis

To provide a structural basis for the binding and signaling of fisetin in suppressing MITF expression, we investigated the crystal structures of PKCα (chain A, PDB ID 3IW4) [56], β-catenin (PDB ID 7AFW) [29], tyrosinase (chain A, PDB ID 2Y9X) [57], and TYRP1 (chain A, PDB ID 5M8M) [58]. The legacy PDB files were obtained from the RCSB Protein Data Bank (RCSB PDB) [59]. All crystallographic water molecules and non-essential coordinates were removed, except for the two Cu ions in tyrosinase and the one water molecule and two Zn ions in the binding site of TYRP1, using UCSF ChimeraX [60,61].

The 3D structure of fisetin was obtained from the PubChem database (CID 5281614) [62] and optimized using the B3LYP/6-31G(d,p) level of theory with Gaussian 09. Docking was performed using the Gnina program (version 1.3.1) [63,64] on a GPU. The binding sites were defined based on the coordinates of the co-crystallized ligands by automatically generating the docking search box with the “-autobox_ligand” option. The exhaustiveness was set to 16, the seed to 0, and all other parameters were kept at their default values. Binding geometries and interactions were visualized using UCSF ChimeraX, v1.10 [60,61].

### 4.13. Statistical Analysis

Data are presented as the mean ± standard deviation (SD) from at least three independent biological experiments. Statistical analysis was performed using one-way ANOVA followed by Tukey’s post hoc test to determine significant differences among groups. All analyses were conducted using GraphPad Prism version 9.0 (GraphPad Software, La Jolla, CA, USA), with *p*-values less than 0.05 considered statistically significant.

## 5. Conclusions

In conclusion, fisetin significantly inhibits melanin production in human melanoma cells by modulating key melanogenic regulators, including MITF and β-catenin. This effect primarily occurs through regulation of PKCα/β-catenin, leading to β-catenin degradation, as well as modulation of the AKT/GSK3β and ERK signaling pathways that induce MITF degradation. These findings suggest that fisetin holds promise as a potential therapeutic or cosmetic agent for treating hyperpigmentation.

## Figures and Tables

**Figure 1 ijms-26-11739-f001:**
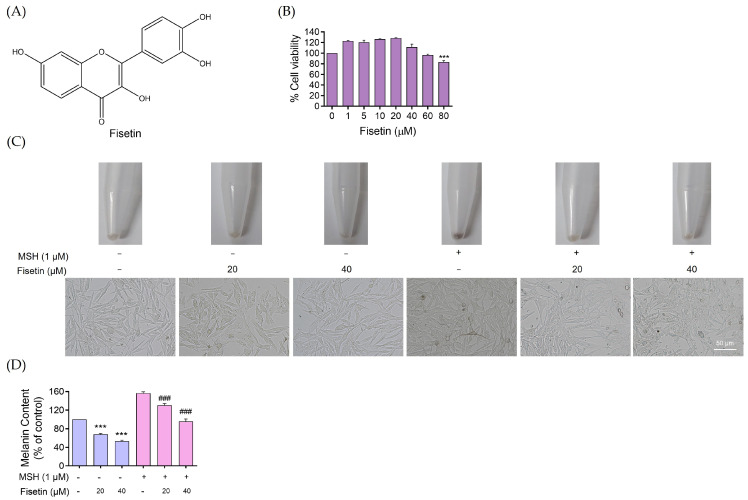
Fisetin reduces melanin synthesis in human melanoma cells. (**A**) Chemical structure of fisetin. (**B**) Human melanoma cells were treated with fisetin (0–80 μM) for 24 h. The cell viability was evaluated by MTT assay. (**C**) The photos show that melanoma cells were pretreated with α-MSH (1 μM) for 24 h prior to fisetin exposure (0–40 μM). (**D**) The α-MSHpretreated melanoma cells were treated with fisetin (0–40 μM). The resulting cell pellets were washed with PBS and then dissolved in 100 μL of 1M NaOH containing 10% DMSO at 80 °C for 3 h. Then, the absorbance was measured at 405 nm. The violet coloration indicated human melanoma cells, while the pink coloration represented α-MSH–pretreated melanoma cells treated with fisetin. Data are presented as mean ± SD (n = 3), with significance indicated as *** *p* < 0.001 compared to untreated control cells and ^###^
*p* < 0.001 compared to α-MSH − treated human melanoma cells.

**Figure 2 ijms-26-11739-f002:**
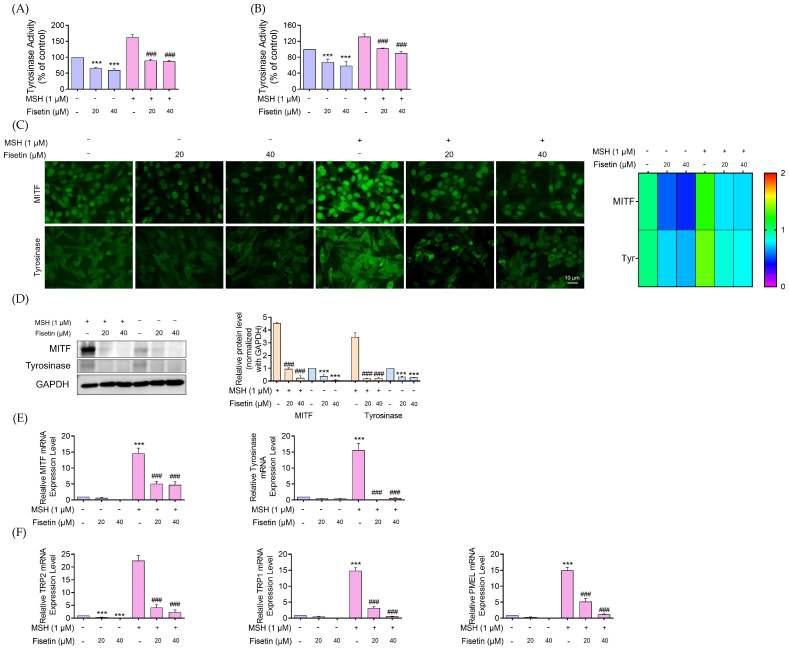
Effect of fisetin on tyrosinase activity, melanogenesis-related protein expression, and melanogenesis-related mRNA level in human melanoma cells. Fisetin (20, 40 μM) was added to human melanoma cells and α-MSH-induced human melanoma cells. (**A**) After treatment, the cells were collected and lysed and cellular tyrosinase activity was measured by using L-Dopa as a substrate. Dopachrome formation was measured via a microplate reader at 490 nm. (**B**) Fisetin and α-MSH-fisetin were directly added at different concentrations to human melanoma cells for cell-free tyrosinase activity assay. The violet coloration indicated human melanoma cells, while the pink coloration represented α-MSH–pretreated melanoma cells treated with fisetin. (**C**) Levels of MITF and tyrosinase were measured through immunofluorescence analysis, with fluorescence intensity quantified using ImageJ software software version 1.52a. (**D**) The protein expression of MITF and Tyrosinase was assessed via Western blotting, with GAPDH used to confirm equal protein loadings. Blots were analyzed by densitometry using ImageJ. The orange coloration indicated α-MSH–pretreated melanoma cells, while the blue coloration represented human melanoma cells treated with fisetin. The uncropped gel image is presented in Appendix A. (**E**) The mRNA expression levels of MITF and tyrosinase. (**F**) The mRNA expression levels of the melanin-related proteins TRP-1, PMEL, and TRP2/DCT were measured. The violet coloration indicated human melanoma cells, while the pink coloration represented α-MSH–pretreated melanoma cells treated with fisetin. The mRNA levels were normalized against the housekeeping gene GAPDH, and relative mRNA expression was calculated using comparative Ct cycles. Data are presented as mean ± SD (n = 3), with significance indicated as *** *p* < 0.001 compared to untreated control cells and ^###^
*p* < 0.001 compared to α-MSH-treated melanoma cells.

**Figure 3 ijms-26-11739-f003:**
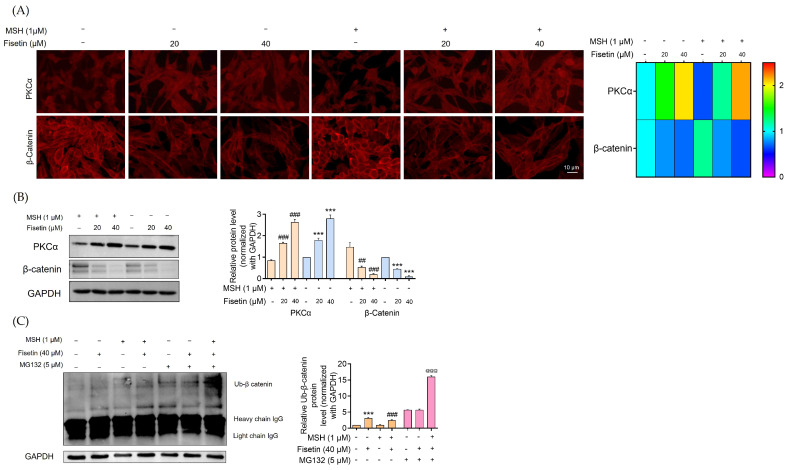
Fisetin reduced melanogenesis activation of the PKCα pathway and induced β-catenin degradation. Melanoma cells and α-MSH-induced melanoma cells were treated with fisetin (20, 40 μM) for 24 h. (**A**) The heat map shows the fluorescence intensity of PKCα activation that led to a reduction in β-catenin captured by a fluorescence microscope; the fluorescence intensity was determined by Image J software version 1.52a. (**B**) The protein expression levels of PKCα and β-catenin were determined by Western blot analysis. The blot was reprobed with GAPDH to confirm the equal loading of proteins. The orange coloration indicated α-MSH–pretreated melanoma cells, while the blue coloration represented human melanoma cells treated with fisetin. The uncropped gel image is presented in Appendix A. (**C**) Fisetin inhibits melanogenesis by a reduction in MITF through β-catenin proteasomal degradation in human melanoma cells and α-MSH-stimulated melanoma cells. Both melanoma and α-MSH-stimulated melanoma cells were exposed to fisetin (40 μM) for 12 h to induce ubiquitin-proteasome-mediated degradation of β-catenin. Prior to fisetin exposure, some cells were pretreated with the proteasome inhibitor MG132 (5 μM) for 1 h. Following treatment, cell lysates were subjected to immunoprecipitation using an anti-β-catenin antibody, and ubiquitinated proteins were detected by Western blotting with an anti-ubiquitin antibody. The level of ubiquitinated β-catenin was then quantified by densitometric analysis. The orange coloration indicated the population without MG132 treatment, whereas the pink coloration represented the population treated with MG132. The uncropped gel image is presented in Appendix A. Data are presented as mean ± SD (n = 3), with significance indicated as *** *p* < 0.001 compared to untreated control cells, ^##^
*p* < 0.01, ^###^
*p* < 0.001 compared to α-MSH-treated melanoma cells, and ^@@@^
*p* < 0.001 compared to MG132-treated melanoma cells.

**Figure 4 ijms-26-11739-f004:**
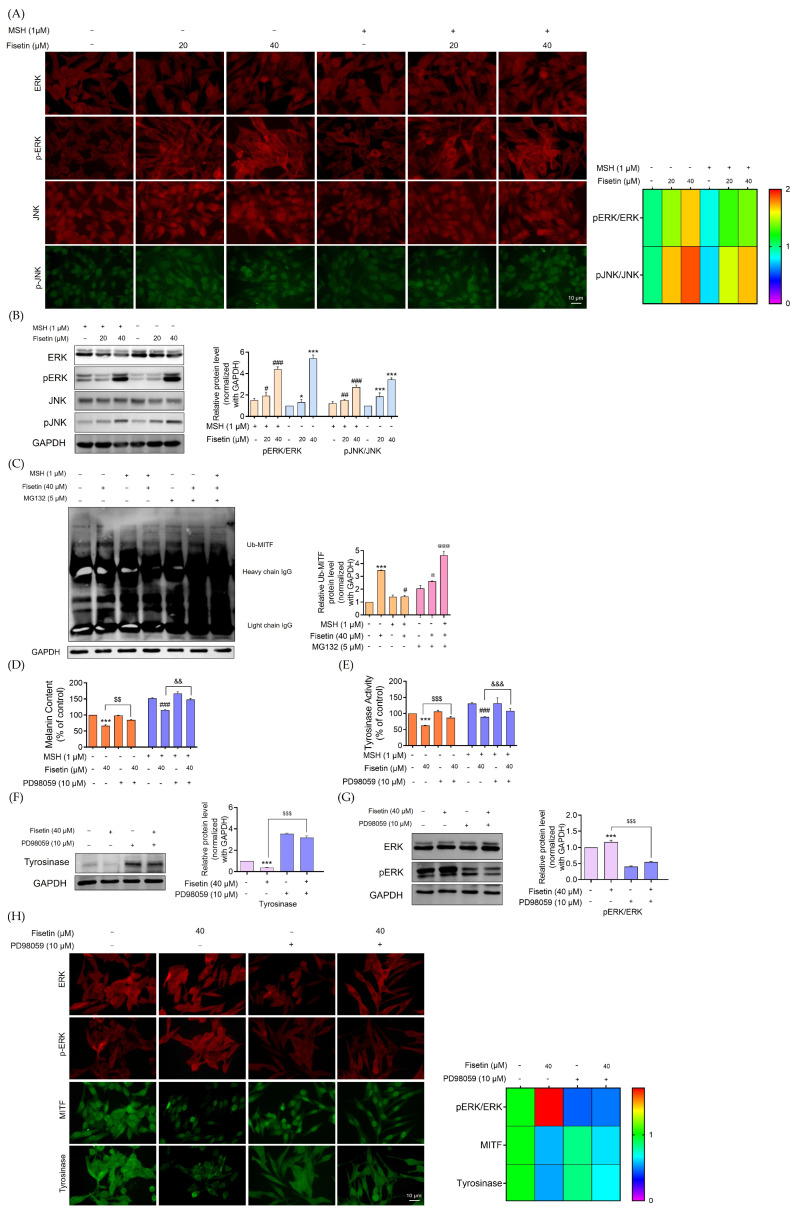
Fisetin reduces melanogenesis via the ERK signaling pathway, leading to MITF degradation. Melanoma cells and α-MSH-induced melanoma cells were treated with fisetin (20, 40 μM) for 24 h. (**A**) The heat map shows the fluorescence intensity of the MITF upstream regulators pERK/ERK and pJNK/JNK captured by a fluorescence microscope; the fluorescence intensity was determined by Image J software version 1.52a. (**B**) The protein expression levels of pERK/ERK and pJNK/JNK were determined by Western blot analysis. The blot was reprobed with GAPDH to confirm the equal loading of proteins. The orange coloration indicated α-MSH–pretreated melanoma cells, while the blue coloration represented human melanoma cells treated with fisetin. The uncropped gel image is presented in Appendix A. (**C**) Fisetin inhibits melanogenesis by a reduction in MITF for proteasomal degradation in human melanoma cells and α-MSH-stimulated melanoma cells. Both melanoma and α-MSH-stimulated melanoma cells were exposed to fisetin (40 μM) for 12 h to induce ubiquitin-proteasome-mediated degradation of MITF. Prior to fisetin exposure, some cells were pretreated with the proteasome inhibitor MG132 (5 μM) for 1 h. Following treatment, cell lysates were subjected to immunoprecipitation using an anti-MITF antibody, and ubiquitinated proteins were detected by Western blotting with an anti-ubiquitin antibody. The level of ubiquitinated MITF was then quantified by densitometric analysis. The orange coloration indicated the population without MG132 treatment, whereas the pink coloration represented the population treated with MG132. The uncropped gel image is presented in Appendix A. (**D**) Human melanoma cells and α-MSH-induced melanoma cells were pretreated with PD98059 (10 μM) for 30 min and fisetin (40 μM) for 24 h to evaluate cellular melanin contents. (**E**) After treatment, the cells were collected and lysed and cellular tyrosinase activity was measured by using L-Dopa as a substrate. Dopachrome formation was measured via a microplate reader at 490 nm. The orange coloration indicated human melanoma cells, while the violet coloration represented α-MSH–pretreated melanoma cells treated with fisetin. (**F**) Human melanoma cells were treated with PD98059 (10 μM) for 30 min, followed by treatment with fisetin (40 μM). The protein expression levels of the melanogenesis enzyme tyrosinase were determined by Western blot analysis. The blot was reprobed with GAPDH to confirm the equal loading of proteins. The uncropped gel image is presented in Appendix A. (**G**) The protein expression levels of pERK/ERK were determined by Western blot analysis. The blot was reprobed with GAPDH to confirm the equal loading of proteins. The pink coloration indicated the population without PD98059 treatment, whereas the violet coloration represented the population treated with PD98059. The uncropped gel image is presented in Appendix A. (**H**) The heat map shows the fluorescence intensity of the MITF upstream regulator pERK/ERK captured by a fluorescence microscope; the fluorescence intensity was determined by Image J software version 1.52a. Data are presented as mean ± SD (n = 3). Significance is shown as * *p* < 0.05 and *** *p* < 0.001 versus untreated control cells, ^#^
*p* < 0.05, ^##^
*p* < 0.01, and ^###^
*p* < 0.001 compared to α-MSH-treated melanoma cells, ^@^
*p* < 0.05 and ^@@@^
*p* < 0.001 compared to MG132-treated melanoma cells, ^&&^
*p* < 0.01 and ^&&&^
*p* < 0.001 versus untreated control cells with PD98059, and ^$$^
*p* < 0.01 and ^$$$^
*p* < 0.001 compared to α-MSH-treated melanoma cells with PD98059.

**Figure 5 ijms-26-11739-f005:**
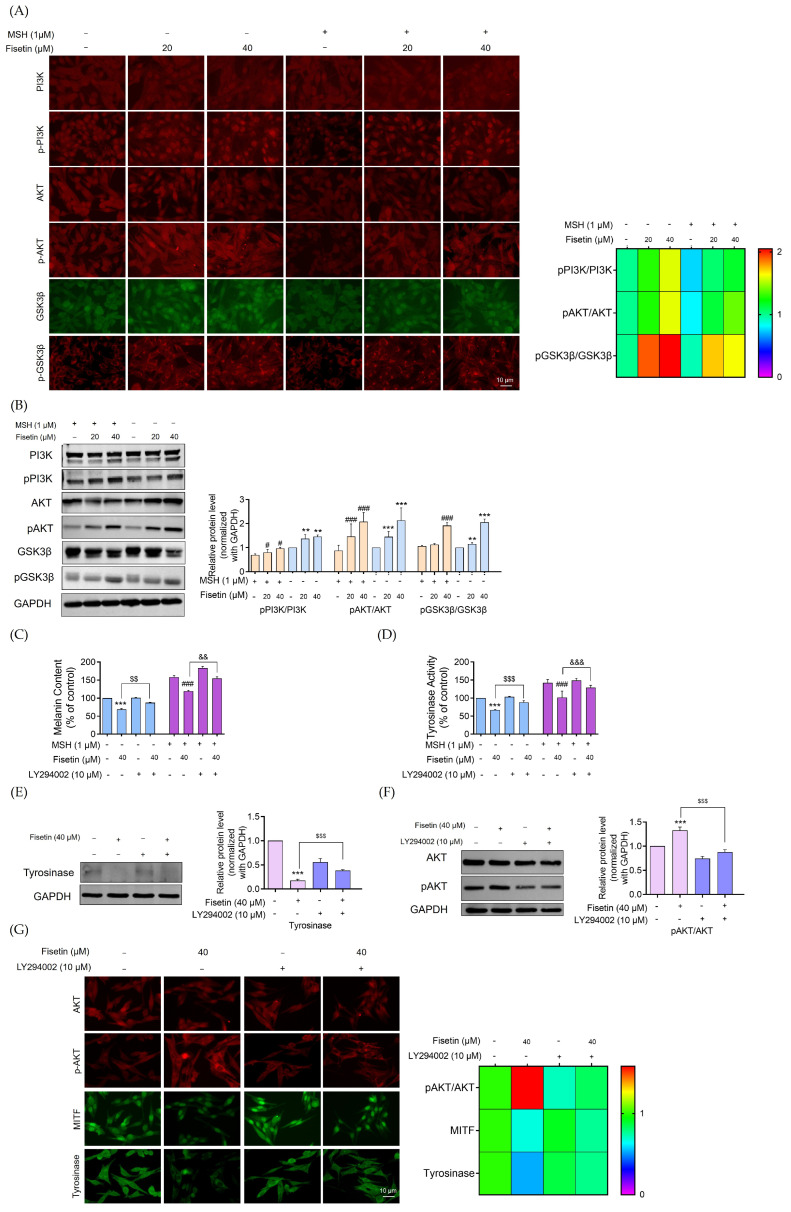
Fisetin reduces melanogenesis via the PI3K/AKT/GSK3β signaling pathway, leading to degradation of MITF. Melanoma cells and α-MSH-induced melanoma cells were treated with fisetin (20, 40 μM) for 24 h. (**A**) The heat map shows the fluorescence intensity of the MITF upstream regulators pPI3K/PI3K, pAKT/AKT, and pGSK3β/GSK3β captured by a fluorescence microscope; the fluorescence intensity was determined by Image J software version 1.52a. (**B**) The protein expression levels of PI3K/AKT/GSK3β were determined by Western blot analysis. The blot was reprobed with GAPDH to confirm the equal loading of proteins. The orange coloration indicated α-MSH–pretreated melanoma cells, while the blue coloration represented human melanoma cells treated with fisetin. The uncropped gel image is presented in Appendix A. (**C**) Human melanoma cells and α-MSH-induced melanoma cells were pretreated with LY294002 (10 μM) for 30 min and fisetin (40 μM) for 24 h to evaluate cellular melanin contents. (**D**) After treatment, the cells were collected and lysed and cellular tyrosinase activity was measured by using L-Dopa as a substrate. Dopachrome formation was measured via a microplate reader at 490 nm. The blue coloration indicated human melanoma cells, while the violet coloration represented α-MSH–pretreated melanoma cells treated with fisetin. (**E**) Human melanoma cells were treated with LY294002 (10 μM) for 30 min, followed by treatment with fisetin (40 μM). The protein expression levels of the melanogenesis protein tyrosinase were determined by Western blot analysis. The blot was reprobed with GAPDH to confirm the equal loading of proteins. The uncropped gel image is presented in Appendix A. (**F**) The protein expression levels of pAKT/AKT were determined by Western blot analysis. The blot was reprobed with GAPDH to confirm the equal loading of proteins. The pink coloration indicated the population without LY294002 treatment, whereas the violet coloration represented the population treated with LY294002. The uncropped gel image is presented in Appendix A. (**G**) The heat map shows the fluorescence intensity of the MITF upstream regulator pAKT/AKT captured by a fluorescence microscope; the fluorescence intensity was determined by Image J software version 1.52a. Data are presented as mean ± SD (n = 3). Significance is shown as ** *p* < 0.01 and *** *p* < 0.001 versus untreated control cells, ^#^
*p* < 0.05 and ^###^
*p* < 0.001 compared to α-MSH-treated melanoma cells, ^&&^
*p* < 0.01 and ^&&&^
*p* < 0.001 versus untreated control cells with LY294002, and ^$$^
*p* < 0.01 and ^$$$^
*p* < 0.001 compared to α-MSH-treated melanoma cells with LY294002.

**Figure 6 ijms-26-11739-f006:**
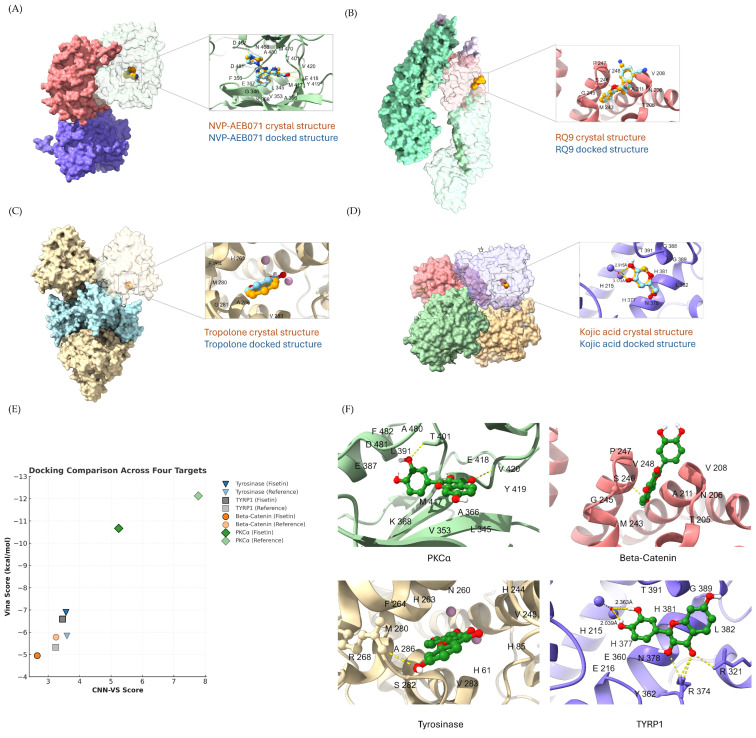
(**A**–**D**) Redocking validation showing close overlap between docked and crystallographic poses for PKCα, β-catenin, tyrosinase, and TYRP1. (**E**) Comparative docking scores of fisetin versus reference inhibitors. (**F**) Predicted binding interactions of fisetin within the binding site.

**Figure 7 ijms-26-11739-f007:**
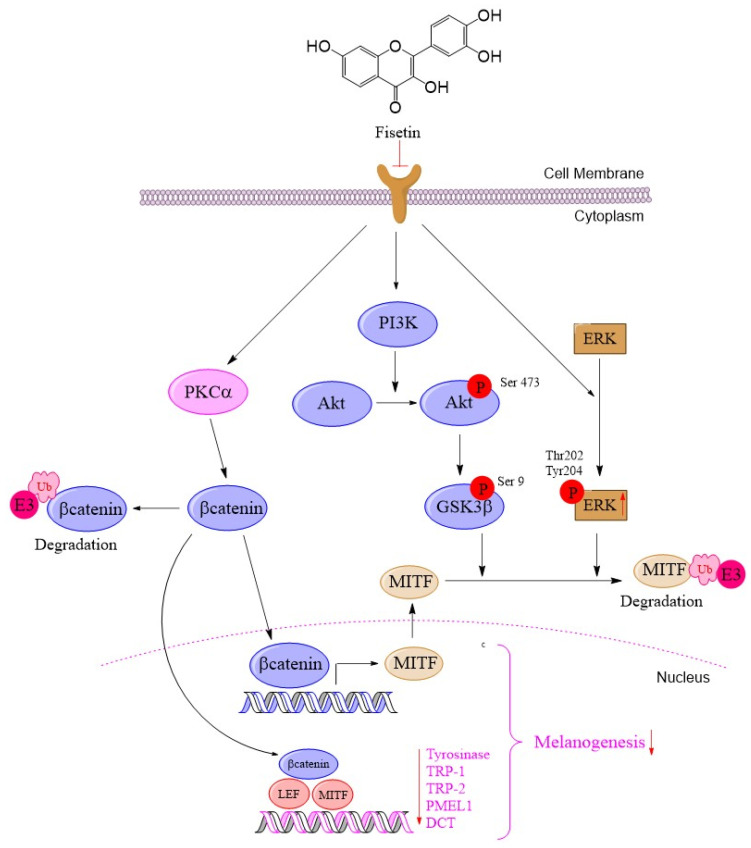
The effect of fisetin suppression of melanogenesis in human melanoma cells in activation of PKCα via degradation of the transcription factor β-catenin. Moreover, activation of phosphorylation of the AKT/GSK3β pathway and the ERK pathway was induced for MITF degradation in human melanoma cells. The MITF reduction in melanogenic-related genes, including tyrosinase, TRP-1, PMEL, and TRP2/DCT, was significantly downregulated in fisetin-treated melanoma cells.

## Data Availability

The original contributions presented in this study are included in the article/Appendix A. Further inquiries can be directed to the corresponding author(s).

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
