# Peer review of "Mechanistic Insights into Anti-Melanogenic Effects of Fisetin: PKCα-Induced β-Catenin Degradation, ERK/MITF Inhibition, and Direct Tyrosinase Suppression"

_ijms, 2025, doi:10.3390/ijms262311739_

Round 1
Reviewer 1 Report
Comments and Suggestions for Authors
This manuscript investigates the multi-target mechanism of fisetin in attenuating melanogenesis, with a focus on PKCα/β-catenin, ERK/MITF signaling pathways, and direct tyrosinase suppression. The study design is well-structured, the experimental data are generally solid, and the conclusions are supported by both in vitro assays and computational analyses. However, several minor issues need clarification or supplementation to enhance the manuscript’s rigor and readability.
2.1 In Section 2.3 (PKCα/β-catenin pathway), the authors show fisetin activates PKCα and promotes β-catenin degradation. However, it remains unclear whether fisetin directly binds to PKCα to induce its activation (as suggested by computational docking in Section 2.6) or indirectly regulates PKCα via upstream signaling molecules. A brief discussion linking the docking results (e.g., ATP-binding pocket interaction) to the observed PKCα activation (e.g., potential allosteric modulation or competitive inhibition of negative regulators) would help bridge this gap.
2.2 In Section 2.4 (ERK pathway), the authors use PD98059 to validate ERK’s role but do not report whether fisetin affects the phosphorylation of other MAPK family members (e.g., p38, JNK) beyond ERK and JNK. While JNK phosphorylation is mentioned, its functional relevance to melanogenesis (e.g., whether JNK inhibition reverses fisetin’s effects) is not addressed. A short supplementary experiment or discussion on JNK’s potential role (or lack thereof) would enhance mechanistic completeness.
Author Response
Reviewer 1
Comments and Suggestions for Authors
This manuscript investigates the multi-target mechanism of fisetin in attenuating melanogenesis, with a focus on PKCα/β-catenin, ERK/MITF signaling pathways, and direct tyrosinase suppression. The study design is well-structured, the experimental data are generally solid, and the conclusions are supported by both in vitro assays and computational analyses. However, several minor issues need clarification or supplementation to enhance the manuscript’s rigor and readability.
2.1 In Section 2.3 (PKCα/β-catenin pathway), the authors show fisetin activates PKCα and promotes β-catenin degradation. However, it remains unclear whether fisetin directly binds to PKCα to induce its activation (as suggested by computational docking in Section 2.6) or indirectly regulates PKCα via upstream signaling molecules. A brief discussion linking the docking results (e.g., ATP-binding pocket interaction) to the observed PKCα activation (e.g., potential allosteric modulation or competitive inhibition of negative regulators) would help bridge this gap.
Response
Thank you for your comment. We agree that clarifying the connection between the computational and experimental findings would strengthen the mechanistic interpretation.
Although the docking results in Section 2.6 (Figure 6) predict that fisetin interacts with the ATP-binding pocket of PKCα (Val420 and Thr401), such binding does not necessarily imply inhibition. Flavonoids are known to modulate PKC activity in a concentration-dependent and context-specific manner rather than functioning as classical inhibitors. For example, Picq [1] et al. (1989) reported a biphasic response in which quercetin stimulates PKC at low concentrations but inhibits it at higher doses [1]. This precedent supports the possibility that fisetin may similarly enhance PKCα activation through mechanisms involving redox regulation, altered membrane microenvironments, or interactions with lipid cofactors, any of which could stabilize or promote PKCα functional activity rather than suppress it [1, 2].
Therefore, even though fisetin interacts with the ATP-binding region, it may promote PKCα activation under specific cellular conditions, leading to β-catenin phosphorylation, its proteasomal degradation, and consequent down-regulation of MITF.
We have revised Section 2.6 (Discussion, page 20 in manuscript) to include this explanation and references accordingly.
References
- Picq M, Dubois M, Munari-Silem Y, Prigent AF, Pacheco H. Flavonoid modulation of protein kinase C activation. Life sciences. 1989;44(21):1563-71.
- Das J, Ramani R, Suraju MO. Polyphenol compounds and PKC signaling. Biochimica et Biophysica Acta (BBA)-General Subjects. 2016;1860(10):2107-21.
2.2 In Section 2.4 (ERK pathway), the authors use PD98059 to validate ERK’s role but do not report whether fisetin affects the phosphorylation of other MAPK family members (e.g., p38, JNK) beyond ERK and JNK. While JNK phosphorylation is mentioned, its functional relevance to melanogenesis (e.g., whether JNK inhibition reverses fisetin’s effects) is not addressed. A short supplementary experiment or discussion on JNK’s potential role (or lack thereof) would enhance mechanistic completeness.
Response
Thank you for your comment. Our findings indicate that activation of the ERK pathway is a key contributor to the fisetin-induced suppression of melanogenesis. In addition to ERK, we also observed increased JNK phosphorylation following fisetin treatment, suggesting that JNK activation participates in the inhibitory response. Although we did not directly test whether pharmacological JNK inhibition can reverse the anti-melanogenic effects of fisetin, prior studies support the notion that JNK activation itself can suppress melanogenesis. Thus, JNK likely plays a supportive or amplifying role in fisetin-mediated melanin reduction, whereas ERK appears to be the dominant signaling axis in our model.
Moreover, because fisetin exerts a multitarget inhibitory mechanism for simultaneously modulating PKCα/β-catenin, ERK/MITF, JNK activation, and directly suppressing tyrosinase-blocking only the JNK pathway may not be sufficient to counteract its overall anti-melanogenic action. Isolating JNK inhibition alone could therefore lead to an incomplete or misleading interpretation of fisetin’s integrated mechanism of action.
This has been revised accordingly in our manuscript.
Introduction
Activation of the JNK signaling pathway is also involved in downregulating the melanin-producing machinery, thereby leading to decreased melanin levels in melanoma cells [1].
Results
Fisetin treatment of melanoma cells resulted in a 2-fold increase in JNK phosphorylation, which promotes MITF degradation. This effect was further confirmed in α-MSH stimulated melanoma cells, where fisetin treatment also enhanced JNK phosphorylation leading to reduce melanin synthesis (Figure 4A).
Discussion
Phosphorylated JNK may play a crucial role in inhibiting melanin production in melanoma cells, likely through negative regulation of the MITF signaling cascade. The selective JNK activator anisomycin suppresses melanogenesis via phosphorylation of CREB-regulated transcription coactivator 3 (CRTC3), whereas JNK inhibition enhances melanin synthesis by promoting CRTC3 dephosphorylation and nuclear translocation. In our findings, fisetin treatment in both melanoma cells and α-MSH-stimulated melanoma cells induces JNK phosphorylation, providing a supportive pathway for MITF degradation and consequent suppression of melanin production [1,2].
References
- Ouyang J, Hu N, Wang H. Petanin potentiated JNK phosphorylation to negatively regulate the ERK/CREB/MITF signaling pathway for anti-melanogenesis in zebrafish. International Journal of Molecular Sciences. 2024;25(11):5939.
- Kim JH, Hong AR, Kim YH, Yoo H, Kang SW, Chang SE, Song Y. JNK suppresses melanogenesis by interfering with CREB-regulated transcription coactivator 3-dependent MITF expression. Theranostics. 2020;10(9):4017.
Reviewer 2 Report
Comments and Suggestions for Authors
The submitted manuscript recapitulates finding published in 2011 by the study "Inhibition of Human Melanoma Cell Growth by the Dietary Flavonoid Fisetin Is Associated with Disruption of Wnt/β-Catenin Signaling and Decreased Mitf Levels". The study is cited by the authors, still the manuscript submitted offers only limited novel findings.
Major points:
1) In Figure 3 PKC activation is shown. Fisetin based induction of PKC is already known, how do the authors envision the mechanism of this activation. Which experiments could provide more insights?
2) Figure 4 is entitled "Effect of Fisetin reduces melanogenesis via ERK signaling pathway led to MITF degrada-268 tion." But in 4E Mek inhibition alone clearly up-regulates TYR, so how can the authors claim that Mek inhibition is Fisetin specific. It rather looks like ageneral mechanisms.
3) The same is true for Fig5E whre LY tretment alone already up-regulated TYR.
4) In Summary Figure 7 GSK3beta is shown to impact MITF, but not to impact beta-catenin. This needs to be corrected since GSK targets beta-catenin for destruction. Additionally, what ubiquitin ligase targets MITF for degredation? Any speculations?
Comments on the Quality of English LanguageNeeds to be improved.
Author Response
Reviewer 2
Comments and Suggestions for Authors
The submitted manuscript recapitulates finding published in 2011 by the study "Inhibition of Human Melanoma Cell Growth by the Dietary Flavonoid Fisetin Is Associated with Disruption of Wnt/β-Catenin Signaling and Decreased Mitf Levels". The study is cited by the authors, still the manuscript submitted offers only limited novel findings.
Major points:
1) In Figure 3 PKC activation is shown. Fisetin based induction of PKC is already known, how do the authors envision the mechanism of this activation. Which experiments could provide more insights?
Response
Thank you for your comment. Protein kinase C alpha (PKCα) has a multifaceted function in the regulation of melanogenesis, the process responsible for melanin production. It can affect melanin synthesis by altering skin pigmentation, and its expression is dynamically regulated in response to various signaling cues associated with pigmentation control [1].
Our findings reveal that fisetin activates PKCα, which in turn suppresses β-catenin levels, leading to enhanced MITF degradation and a consequent reduction in melanin and tyrosinase production in human melanoma cells. This discovery establishes a novel link between PKCα activation and the inhibition of melanogenesis by fisetin-an association that, to our knowledge, has not been previously reported.
Additionally, we observed that other compounds can also modulate PKC-α activity to regulate melanogenesis in melanoma cells. Retinoic acid (RA) has been reported to suppress melanogenesis in B16 mouse melanoma (B16) cells. Conversely, RA has also been found to enhance protein kinase C (PKC) activity in these cells. To clarify this relationship, further studies were conducted to determine which PKC isoforms are expressed in B16 cells and how their expression levels are affected by RA treatment. The findings revealed that RA selectively increases the α isoform of PKC, leading to the inhibition of melanogenesis in melanoma cells [2, 3].
Overall, our study presents a novel finding that fisetin suppresses melanogenesis in both melanoma cells and α-MSH-stimulated melanoma cells by activating PKC-α, which promotes MITF degradation and consequently reduces melanin synthesis.
References
- Mahalingam H, Vaughn J, Novotny J, Gruber JR, Niles RM. Regulation of melanogenesis in B16 mouse melanoma cells by protein kinase C. Journal of cellular physiology. 1996;168(3):549-58.
- Oka M, Ogita K, Saito N, Mishima Y. Selective increase of the α subspecies of protein kinase C and inhibition of melanogenesis induced by retinoic acid in melanoma cells. Journal of investigative dermatology. 1993;100(2):S204-8.
- Bertolotto C, Bille K, Ortonne JP, et al. In B16 melanoma cells, the inhibition of melanogenesis by TPA results from PKC activation and diminution of microphthalmia binding to the M-box of the tyrosinase promoter. Oncogene. 1998; 16:1665-1670.
2) Figure 4 is entitled "Effect of Fisetin reduces melanogenesis via ERK signaling pathway led to MITF degradation." But in 4E Mek inhibition alone clearly up-regulates TYR, so how can the authors claim that Mek inhibition is Fisetin specific. It rather looks like a general mechanisms.
Response
Thank you for your comment. Our findings showed that fisetin treatment alone reduced tyrosinase activity in melanoma cells to approximately 45% of the level observed in untreated controls (100%). However, when cells were treated with the ERK/MEK inhibitor PD98059, tyrosinase activity decreased by only about 10% compared with fisetin-treated cells. These findings indicate that blocking the ERK/MEK pathway leads to increased MITF accumulation, which in turn elevates tyrosinase activity compared to fisetin treatment alone. Overall, the results indicate that the ERK/MEK signaling pathway plays a key regulatory role in melanin synthesis in fisetin-treated melanoma cells.
3) The same is true for Fig5E whre LY tretment alone already up-regulated TYR.
Response
Thank you for your comment. Our findings showed that fisetin treatment alone reduced tyrosinase activity in melanoma cells to approximately 35% of the level observed in untreated controls (100%). In contrast, treatment with the AKT inhibitor LY294002 resulted in only about a 10% decrease in tyrosinase activity compared to fisetin-treated cells. These findings imply that suppressing the AKT pathway leads to increased MITF accumulation, which consequently elevates tyrosinase activity compared to fisetin treatment alone. Overall, this indicates that the AKT signaling pathway plays a crucial role in regulating melanin synthesis in fisetin-treated melanoma cells.
4) In Summary Figure 7 GSK3beta is shown to impact MITF, but not to impact beta-catenin. This needs to be corrected since GSK targets beta-catenin for destruction. Additionally, what ubiquitin ligase targets MITF for degredation? Any speculations?
Response
Thank you for your comment. Ubiquitin is a highly conserved polypeptide that plays a critical role in the ubiquitin-proteasome system. In ubiquitination, ubiquitin molecules are covalently linked to target proteins, marking them for degradation by the 26S proteasome. This process occurs through three main enzymatic steps: ubiquitin is first activated by forming a thioester bond with the E1 activating enzyme, then transferred to the E2 conjugating enzyme, and finally, the E3 ubiquitin ligase facilitates the attachment of ubiquitin to specific lysine residues on the substrate protein [1].
MITF is frequently subjected to ubiquitination, leading to its proteasomal degradation. For instance, the ubiquitin-conjugating enzyme hUBC9 interacts directly with MITF, promoting its breakdown. This regulation ensures proper MITF levels in the cell. hUBC9 also influences melanocyte differentiation by mediating MITF sumoylation at Lys182 and Lys316 and facilitating its degradation. E3 ligases play a key role in controlling MITF protein stability by targeting it for proteasomal degradation [2-4].
Similarly, β-catenin undergoes degradation through the ubiquitin-proteasome pathway. Its ubiquitination also involves the sequential action of E1, E2, and E3 enzymes. Phosphorylated β-catenin is recognized by E3 ligase complexes, such as those containing β-TrCP, which attach ubiquitin chains to signal its degradation [5].
Our findings indicate that phosphorylated GSK3β at Ser9 does not affect β-catenin degradation in melanoma cells treated with fisetin. However, Activation of the PI3K/Akt pathway promotes GSK3β phosphorylation at Ser9, which ultimately suppresses MITF expression [6, 7]. Our novel findings indicate that activation of PKCα serves as the key regulator responsible for suppressing β-catenin in fisetin-treated human melanoma cells.
References
- Park HB, Kim JW, Baek KH. Regulation of Wnt signaling through ubiquitination and deubiquitination in cancers. International journal of molecular sciences. 2020;21(11):3904.
- Xu W, Gong L, Haddad MM, Bischof O, Campisi J, Yeh ET, Medrano EE. Regulation of microphthalmia-associated transcription factor MITF protein levels by association with the ubiquitin-conjugating enzyme hUBC9. Experimental cell research. 2000;255(2):135-43.
- Hu S, Bai S, Dai Y, Yang N, Li J, Zhang X, Wang F, Zhao B, Bao G, Chen Y, Wu X. Deubiquitination of MITF-M regulates melanocytes proliferation and apoptosis. Frontiers in Molecular Biosciences. 2021;8:692724.
- Xu W, Gong L, Haddad MM, Bischof O, Campisi J, Yeh ET, Medrano EE. Regulation of microphthalmia-associated transcription factor MITF protein levels by association with the ubiquitin-conjugating enzyme hUBC9. Experimental cell research. 2000;255(2):135-43.
- Liu C, Kato Y, Zhang Z, Do VM, Yankner BA, He X. β-Trcp couples β-catenin phosphorylation-degradation and regulates Xenopus axis formation. Proceedings of the National Academy of Sciences. 1999;96(11):6273-8.
- Ngeow KC, Friedrichsen HJ, Li L, Zeng Z, Andrews S, Volpon L, Brunsdon H, Berridge G, Picaud S, Fischer R, Lisle R. BRAF/MAPK and GSK3 signaling converges to control MITF nuclear export. Proceedings of the National Academy of Sciences. 2018;115(37):E8668-77.
- Choi H, Yoon JH, Youn K, Jun M. Decursin prevents melanogenesis by suppressing MITF expression through the regulation of PKA/CREB, MAPKs, and PI3K/Akt/GSK-3β cascades. Biomedicine & Pharmacotherapy. 2022;147:112651.
Comments on the Quality of English Language
Needs to be improved.
Response
Thank you for your comment. We have consulted a language specialist regarding our manuscript and have incorporated the suggested revisions.

Round 2
Reviewer 2 Report
Comments and Suggestions for Authors
The questions raised were commented sufficiently.
Author Response
Comments and Suggestions for Authors
The questions raised were commented sufficiently.
Response: Thank you very much for your comments. The manuscript has been thoroughly checked and revised by a language specialist as recommended.
